# Foegraecumoside O and P, a Pair of Triterpenoid Saponins with a 4/5/6 Fused Tricyclic Oleanane Carbon Skeleton from *Lysimachia foenum-graecum* Hance

**DOI:** 10.3390/molecules28135061

**Published:** 2023-06-28

**Authors:** Lumei Dai, Shuang He, Bin Zhang, Hengshan Wang, Yan Wang, Dong Liang

**Affiliations:** 1State Key Laboratory for Chemistry and Molecular Engineering of Medicinal Resources, School of Chemistry and Pharmaceutical Sciences, Guangxi Normal University, Guilin 541004, China; dlmei610@163.com (L.D.); m15077301367@163.com (S.H.); zhbin308@163.com (B.Z.); whengshan@163.com (H.W.); 2School of Biological and Food Engineering, Huanghuai University, Zhumadian 463000, China; 3H. E. J. Research Institute of Chemistry, International Center for Chemical and Biological Sciences, University of Karachi, Karachi 75270, Pakistan

**Keywords:** Primulaceae, *Lysimachia*, *Lysimachia foenum-graecum*, triterpenoids, cytotoxicity

## Abstract

*Lysimachia foenum-graecum* Hance (Primulaceae) is a medicinal plant used for cold, pain, ascariasis, etc., in China. Triterpenoid saponins have been found to be the main components of this genus. In this work, a pair of oleanane-type triterpenoid saponins with an unprecedented 4/5/6 fused tricyclic skeleton, foegraecumoside O (**1**) and foegraecumoside P (**2**) were isolated from the butanol fraction of the aerial parts of *L. foenum-graecum*. Their structures were determined using chemical methods and extensive spectroscopic analyses, along with quantum chemical calculations. Compound **2** displayed moderate cytotoxicity against HepG2, MGC-803, T24, NCI-H460, A549, and A549/CDDP (drug-resistant lung-cancer cell line) with IC_50_ at 12.4–19.2 μM in an MTT assay, comparing with the positive control doxorubicin, which had IC_50_ at 0.53–4.92 μM, but was inactive for A549/CDDP. Furthermore, a possible biosynthetic pathway for forming compounds **1** and **2** was proposed.

## 1. Introduction

There is a long history of human beings fighting with cancer [1]. After the development and application of multiple generations of anticancer drugs, drug resistance was gradually produced, due to which the effects of anticancer drugs declined progressively. Therefore, there is always need to develop new potential therapeutic molecules, especially those molecules that are active against drug-resistant cancers [2,3].

The *Lysimachia* genus (Primulaceae) comprises approximately 180 species that are widespread in temperate and subtropical regions, 138 of which can be found in China [4]. *Lysimachia foenum-graecum* Hance is a species that is distributed mainly in the Guangxi Zhuang Autonomous Region and the Guangdong, Hunan, and Yunnan Provinces of China [4]. It is famous due to its pleasant smell after drying; therefore, it is commonly used as perfumery material. The aerial parts of this plant have been used for the treatment of colds, headaches, sore throats, toothaches, abdominal pain, ascariasis, and other diseases in traditional Chinese medicine [5]. Previous studies indicated that triterpenoid saponins are the main chemical constituents of this genus, and many of them were found to be cytotoxic [6,7,8,9,10,11,12]. Among them, our group has contributed to the discoveries of new components from *L. clethroides*, *L. fortune*, and *L. foenum-graecum* [8,10,11,12]. In previous phytochemistry investigation, many triterpenoid saponins and a few flavonoids have been isolated from *L. foenum-graecum* [10,12,13,14,15,16,17]. Some reported saponins were found to possess a cytotoxic property against cancer cell lines, including NCI-H460, MGC-803, HepG2, T24, A549, and A549/CDDP. [10,12,13].

In terms of exploring the more active natural molecules from this genus, a pair of novel compounds, foegraecumoside O (**1**) and foegraecumoside P (**2**), featuring unique 4/5/6 fused tricyclic skeletons linked between C-18 and C-30, were purified and identified from *L. foenum-graecu* (Figure 1). Based on the presence of the chiral center at C-30, which makes the two compounds different, quantum chemical calculations were performed to simulate the theoretical NMR data of the optical isomers. Thereafter, the absolute configurations of the isomers were able to be determined. Herein, the isolation, structural characterization, inhibitory effect on the growth of tumor cells, and proposed biosynthesis of these two molecules were discussed.

## 2. Results

### 2.1. Structure Elucidation

The aerial parts of *L*. *foenum-graecum* were extracted by 95% EtOH. Then, the obtained crude was suspended in water and extracted with EtOAc and *n*-BuOH, successively. Two compounds, **1** and **2** (Figure 1), were isolated through multiple chromatography columns from the BuOH fraction and then purified using reverse-phase HPLC.

As a white, amorphous powder, compound **1** owned a negative optical rotation in MeOH, [α${]}_{D}^{20}$ − 11.3 (*c* 0.1, MeOH). Its molecular formula was determined to be C_53_H_86_O_22_ via HR-ESI-MS (*m/z* 1097.5496 [M + Na]^+^, calculated for 1097.5503) (Appendix A). Absorbances (cm^−1^) at 3436 (OH), 1075 (C-O-C), and 2940 (CH_3_, CH_2_, and CH) were noted in the IR spectrum. In the ^1^H NMR spectrum (pyridine-d_5_, 500 MHz) (Table 1, Appendix A), six tertiary methyls at *δ*_H_ 0.80, 0.98, 1.13, 1.23, 1.38, and 1.50, and a pair of geminal protons at *δ*_H_ 3.83 and 3.69 (each 1H, d, *J* = 8.0 Hz) were observed, which correlated with six methyl carbon resonances at *δ*_C_ 16.3, 16.4, 28.0, 21.3, 18.3, and 19.1, respectively, and an oxygenated methylene at *δ*_C_ 76.4 in HSQC (Appendix A). In addition, a quaternary carbon resonance occurred at *δ*_C_ 88.9 in ^13^C NMR (Appendix A). All these data were similar to 13,28-epoxyoleanane skeleton, except for the absence of a tertiary methyl [8]. An oxymethine, with a carbon at *δ*_C_ 81.0 and correlated with a proton at *δ*_H_ 4.55 in HSQC, was assigned to C-30 by the HMBC correlation of *δ*_H_ 4.55 with C-17 (*δ*_C_ 49.6), and correlations of *δ*_C_ 81.0 with H_3_-29 (*δ*_H_ 1.23), Ha-19 (3.38), and Ha-21 (2.37), indicative of the presence of a 4/5/6 fused tricyclic skeleton formed through the linkage of C-18 and C-30 (Figure 2). Moreover, two other oxymethine protons were observed at *δ*_H_ 3.10 (dd, *J* = 11.5, 4.0 Hz) and 4.15 (overlapped). The peak at *δ*_H_ 3.10 was confirmed to be located at H-3 through the heteronuclear correlations of its carbon at *δ*_C_ 89.1 with H_3_-23 (*δ*_H_ 1.13) and H_3_-24 (*δ*_H_ 0.98). Another proton at *δ*_H_ 4.15 was assigned to H-16 because of the COSY correlations with H-15 (*δ*_H_ 2.33 and 1.55) and the HMBC correlations with C-14 (*δ*_C_ 46.1) and C-17 (*δ*_C_ 49.6) (Appendix A). In the NOESY spectrum, the cross peaks of H-3 (*δ*_H_ 3.10)/H-5 (*δ*_H_ 0.60) indicated the α-orientation of H-3, whereas the cross peaks of H-16 (*δ*_H_ 4.15) and H-28b (*δ*_H_ 3.69) confirmed the β-orientation of H-16. Moreover, the NOESY correlation between H-30 (*δ*_H_ 4.55) and H-28a (*δ*_H_ 3.83) indicated the configuration of H-30 as α-orientated (Figure 2 and Appendix A). Accordingly, the aglycone of compound **1** was defined as 3β,16α,30β-trihydroxy-13β, 28-epoxy-18,30-cyclo-oleanane.

The ^1^H NMR of **1** exhibited signals of four sugar anomeric protons at *δ*_H_ 4.94 (1H, br s), 5.34 (1H, d, *J* = 8.0 Hz), 5.21 (1H, d, *J* = 7.5 Hz), and 6.39 (1H, br s), which correlated to four anomeric carbons at *δ*_C_ 104.3, 105.4, 103.1, and 101.5, respectively, in the HSQC spectrum. After acid hydrolysis, derivatization of standard sugars and products, and HPLC analysis with an optical detector, the sugar moieties were proved to be L-arabinose, D-glucose, and L-rhamnose with a ratio of 1:2:1 [10,18]. The HMBC spectrum exhibited the correlation between H-3 (*δ*_H_ 3.10) with Ara-C-1 (*δ*_C_ 104.3). Ara-H-2 (*δ*_H_ 4.54) correlated with Glc-C-1 (*δ*_C_ 105.4). Although long-range heteronuclear coupling between Ara and Glc-II is too weak to observe in HMBC, the cross peak of Glc-II-H-I (*δ*_H_ 5.21) and Ara-H-4 (*δ*_H_ 4.56) was observed in NOESY, which can also prove the location of Glc-II. Furthermore, Glc-II-H-2 (*δ*_H_ 4.26) showed a correlation with Rha-C-1 (*δ*_C_ 101.5) in HMBC. Thereby, the sequence of the sugar chain composed of the four sugar units was determined as shown in Figure 1, which is the same as what appeared in the previous reports [8,10,11]. The relative configurations of anomeric protons of two glucoses at *δ*_H_ 5.34 (Glc I) and 5.21 (Glc II) were determined as β based on their coupling constants (8.0 Hz and 7.5 Hz, respectively). The arabinose unit was identified to be the α-anomer according to the correlations of Ara-H-1 (*δ*_H_ 4.94) with Ara-H-3 (*δ*_H_ 4.49) and Ara-H-5b (*δ*_H_ 3.78) respectively in the NOESY spectrum. Furthermore, Rha-H-1 (*δ*_H_ 6.39) correlated with Rha-H-2 (*δ*_H_ 4.71) and confirmed the α-anomeric orientation of the rhamnopyranose unit. Hence, compound **1** was elucidated as shown in Figure 1.

Compound **2**, with a negative optical rotation in MeOH, [α${]}_{D}^{20}$ − 11.3 (*c* 0.1, MeOH), was purified as a white amorphous powder. It was deduced to have an identical molecular formula to that of compound **1** according to its HR-ESI-MS data (Appendix A). The IR spectrum displayed absorptions at 3426 (OH), 2940 (CH_3_, CH_2_, and CH), at 1075 (C-O-C) cm^−1^. The ^1^H and ^13^C NMR data of compound **2** closely resembled those of compound **1** (Table 1, Appendix A), indicating that these two compounds are structurally similar. Comprehensive comparisons of their 2D NMR spectra (Appendix A) suggested that the structure of compound **2** is practically same to that of compound **1**, except for the configuration of H-30, which was confirmed as β-orientated through the correlation of H-30 (*δ*_H_ 3.96) with H_2_-12 (*δ*_H_ 1.73, 1.85) and H-29 (*δ*_H_ 1.18) observed in the NOESY spectrum of compound **2** (Figure 3). Thus, compound **2** was proved as shown in Figure 1. NMR data of compounds **1** and **2** are assigned in Table 1.

### 2.2. NMR Calculation

To further verify the configuration of the hydroxy at C-30, a quantum chemical calculation of the ^1^H NMR and ^13^C NMR chemical shifts for the aglycones of compounds **1** and **2** was conducted using the gauge-including atomic orbitals (GIAO) method at the mPW1PW91/6-311+G (2d, p) in pyridine with the IEFPCM model [19,20]. As shown in Figure 4, as well as Appendix A, the experimental ^13^C NMR values of the aglycone of compound **1** has an R^2^ (coefficient of determination) of 0.9917 with theoretical values of the aglycone of compound **1** (Figure 4A) but had one of 0.9845 with the aglycone of compound **2** (Figure 4B). Similarly, the experimental ^13^C NMR values of the aglycone of compound **2** had an R^2^ of 0.9914 with theoretical values of the aglycone of compound **2** (Figure 4D) but had one of 0.9818 with the aglycone of **1** (Figure 4C). The linear regression fitting of ^1^H NMR (Figure 5, Appendix A) displayed that the experimental ^1^H NMR data of the aglycone of compound **1** owns an R^2^ of 0.9451 (Figure 5A) but had one of 0.8604 with the aglycone of compound **2** (Figure 5B). Likewise, the experimental ^1^H NMR values of the aglycone of compound **2** had an R^2^ of 0.9588, with theoretical values of the aglycone of compound **2** (Figure 5D) but had one of 0.8719 with the aglycone of compound **1** (Figure 5C). It is therefore concluded that the calculated ^1^H NMR and ^13^C NMR chemical shifts for the aglycones of compounds **1** and **2** showed a better agreement with the experimental values with a higher correlation coefficient. Consequently, structures of compounds **1** and **2** were further confirmed.

### 2.3. Cytotoxicity Assay

Compounds **1** and **2** were evaluated for their cytotoxic activities against five human cancer cell lines in vitro using the MTT method with doxorubicin as the positive control (Table 2). Compound **2** showed moderate cytotoxicities against NCI-H460, MGC-803, HepG2, and T24, with IC_50_ values of 18.4, 12.4, 19.2, and 15.0 μM, respectively. Furthermore, compound **2** was tested on drug-sensitive and drug-resistant lung-cancer cell lines (A549 and A549/CDDP, respectively), and it displayed moderate cytotoxicity against A549/CDDP, with an IC_50_ value of 16.0 μM and a resistance factor (RF) of 0.94 (Table 3).

### 2.4. Biosynthetic Pathway

The biosynthetic pathway of compounds **1** and **2** was postulated as shown in Figure 1. Firstly, the five-carbon building blocks, 3-isopentenyl pyrophosphate (IPP) and dimethylallyl pyrophosphate (DMAPP), were synthesized using the 2-C-methyl-D-erythritol 4-phosphate (MEP) or mevalonic acid (MVA) pathways [21]. Then, six building blocks were condensed to form C_30_ squalene, which is the precursor of all triterpenoids in eukaryotes. Subsequently, the linear squalene was epoxidized to 2,3-oxidosqualene. The 2,3-oxidosqualene was cyclized to the pentacyclic oleanane-type triterpenoid backbone β-amyrin by the β-amyrin synthase (β-AS). Furtherly oxidization of the β-amyrin by CYP450s introduced an oxygen atom into the specific site of their substrates to primarily form hydroxyl, carboxyl, or epoxy groups [22]. Finally, these molecules were glycosylated to triterpenoid saponins via UDP-dependent glycosyltransferase (UGTs) [23].

## 3. Discussion

Triterpenoids and their saponins play an important role in natural product chemistry. Among them, the oleanane-type skeleton is a classical pentacyclic skeleton and commonly exists in many families, such as Araliaceae [24], Fabaceae [25], Campanulaceae [26], Polygalaceae [27], etc. In the previous phytochemical investigation of the Primulaceae family, it was found that oleanane-type skeletons, especially 13,28-epoxyoleanane skeletons, were the general feature of the main components of this family [28]. As a genus from the Primulaceae family, many oleanane-type triterpenoid saponins, more than half of which having beard 13,28-epoxyoleanane skeletons, have been isolated from the *Lysimachia* genus [6,7,8,9,10,11,12,13,14,15]. In this work, two epimers featuring a unique 4/5/6 fused tricyclic skeleton linked between C-18 and C-30 were isolated from *L. foenum-graecu*. This is the first example of this novel skeleton within the large family of oleanane-type triterpenoids. Therefore, it is an important discovery for the chemical diversity of triterpenoids.

As a pair of epimers with different configurations of 30-OH, compounds **1** and **2** displayed high stereoselectivity against cancer cell lines. Compound 2, with 30α-OH, was active against several cancer cell lines, including the drug-resistant lung-cancer cell line A549/CDDP. Whereas compound **1** with 30α-OH was proven to be inactive against all those tested cell lines. Meanwhile, with an RF of 0.94, compound 2 displayed a similar inhibitory effect against normal lung-cancer cells and drug-resistant lung-cancer cells, showing its potential to be a new approach for drug resistance. The results provide a potential lead compound for further investigation in treating diseases related to cancer and drug resistance to cancer.

## 4. Materials and Methods

### 4.1. General Experimental Procedures

A PerkinElmer model 341 polarimeter was used to record optical rotations. A Thermo-Scientific Exactive mass spectrometer was used to carry out HR-ESI-MS spectrum. A PerkinElmer Spectrum Two FT-IR spectrometer was used to obtain IR spectra. The 1D and 2D NMR spectra were performed on Bruker AV-500 or AV-600 MHz spectrometers in deuterated pyridine. Chemical shifts (*δ*) were reported in ppm relative to the solvent signals, and coupling constants (*J*) were calculated and reported in Hz. A Shimadzu LC-6AD HPLC with an RID-10A detector and an YMC-Pack ODS-A column (250 mm × 20 mm, 5 μm) was used to perform purification of two compounds. A Jasco LC-4000 Analytical HPLC was used to analyze extracts, fractions, and semipure samples. Different materials, including silica gel (200–300 mesh, Qingdao Marine Chemical Factory, Qingdao, China), MCI gel (CHP20, 75–150 μm, Mitsubishi Chemical Corporation, Tokyo, Japan), and ODS (50 μm, YMC, Kyoto, Japan), were used for column chromatography (CC) for isolation of pure compounds. Precoated silica gel GF254 plates (Qingdao Marine Chemical Factory) were used for TLC analysis. Spots were detected on TLC by observing them under UV light and then heating after spraying with 10% H_2_SO_4_ in EtOH. Solvents used for extraction and chromatography column were all analytical grade and manufactured by Xilong Scientific, Shantou, China. HPLC-grade methanol and acetonitrile were manufactured by Tedia, Plzen, Czech Republic.

### 4.2. Plant Material

The aerial parts of *L. foenum-graecum* were collected in Jinxiu, Guangxi Zhuang Autonomous Region, People’s Republic of China. Shaoqing Tang from Guangxi Normal University authenticated the species. A voucher specimen (NO. LF-2014022) was deposited in the archive of the State Key Laboratory for Chemistry and Molecular Engineering of Medicinal Resources, School of Chemistry and Pharmaceutical Sciences, Guangxi Normal University, Guilin, China.

### 4.3. Extraction and Isolation

The dry aerial parts of *L*. *foenum-graecum* (14.7 kg) were extracted with 95% aqueous EtOH at reflux (3 × 3 h × 75 L). After evaporation under a vacuum (45 °C in water bath), the obtained crude extract (1.6 kg) was suspended in water and extracted with EtOAc and *n*-BuOH, successively. The *n*-BuOH fraction was subjected to microporous resin CC eluting with aqueous ethanol (20%, 50%, 70%, and 95%, *v/v*). Subsequently, the fraction from 70% EtOH (228 g) was fractionated with silica gel CC eluting with a gradient of CH_2_Cl_2_-MeOH (10:1→0:1, *v/v*) to afford six fractions (A–F). Fraction E (92.8 g) was further separated by MCI gel CC with a MeOH-H_2_O gradient elution system (50:50→100:0, *v/v*) to obtain 10 subfractions (E1–E10). E6 (3.5 g) was chromatographed over RP-C_18_ CC eluting with MeOH:H_2_O (50:50→65:35, *v/v*, then MeOH) to yield E6-1 to E6-11. E6-7 (142.7 mg) was applied on a preparative HPLC (CH_3_CN-H_2_O, 29:71, *v/v*, at 8 mL/min) to obtain compounds **1** (16.8 mg, 38.1 min) and **2** (4.0 mg, 31.0 min).

Foegraecumoside O (1): Amorphous powder, ${\left[\alpha \right]}_{D}^{20}$ − 11.3 (*c* 0.1, MeOH); IR (KBr) *ν*_max_ 3436, 2940, 1637, 1449, 1393, 1075, 606 cm^−1^; ^1^H NMR and ^13^C NMR data, see Table 1; HR-ESI-MS *m/z* 1097.5496 [M + Na]^+^ (calculated for C_53_H_86_O_22_Na, 1097.5508).

Foegraecumoside P (2): Amorphous powder, ${\left[\alpha \right]}_{D}^{20}$ − 11.3 (*c* 0.1, MeOH); IR (KBr) *ν*_max_ 3426, 2940, 1637, 1393, 1075, 615 cm^−1^; ^1^H NMR and ^13^C NMR data, see Table 1; HR-ESI-MS *m/z* 1097.5493 [M + Na]^+^ (calculated for C_53_H_86_O_22_Na, 1097.5508).

### 4.4. Acid Hydrolysis

Each compound (2 mg) was dissolved in 1 M HCl (1,4-dioxane-H_2_O, 1:1, 5 mL; Xilong Scientific, Shantou, China) and then stirred at 80 °C for 8 h. The reaction mixture was extracted with CH_2_Cl_2_ (Xilong Scientific, Shantou, China) after cooling. Afterward, each aqueous layer was evaporated under a vacuum (50 °C in water bath), and then diluted with H_2_O for multiple times to provide a neutral residue. Each residue was subjected to analytical HPLC (Jasco LC-4000, Tokyo, Japan) under the following conditions: Shodex Asahipak NH2P-50 4E column (250 mm × 4.6 mm, 5 mm); Jasco OR-4090 optical rotation detector; mobile phase, CH_3_CN:H_2_O (78:22, *v/v*); flow rate 1 mL/min. The absolute configurations of sugar units in compounds **1** and **2**, composed of glucose, arabinose, and rhamnose, were confirmed by comparing their retention times and optical rotations with those of authentic samples (National Institute for Food and Drug Control, Beijing, China) [18]. Authentic sugars had retention times of *t*_R_: 6.5 min (l-rhamnose, negative optical rotation), 7.7 min (l-arabinose, positive optical rotation), and 11.3 min (d-glucose, positive optical rotation).

### 4.5. Quantum Chemical Calculations

NMR calculations were carried out via Gaussian 09, following the protocol adapted from Michael et al. [29]. At first, structures were optimized at B3LYP/6-31+G(d,p) theory level in gas phase. Then, the NMR calculations were conducted using the gauge-including atomic orbitals (GIAO) method at mPW1PW91/6-311+G (2d, p) in pyridine using the IEFPCM model. Finally, the TMS-corrected NMR chemical-shift values were fitted to the experimental values using the ordinary least-squares linear-regression (OLSLR) method. The calculated ^13^C NMR and ^1^H NMR chemical-shift values of TMS in pyridine were 187.32 ppm and 31.73 ppm, respectively.

### 4.6. Cytotoxicity Assay

Cytotoxicities of compounds **1** and **2** were tested using the 3-(4,5-dimethylthiazol-2-yl)-2,5-diphenyltetrazolium bromide (MTT) method, as described using human non-small-cell lung carcinoma (NCI-H460), human gastric carcinoma (MGC-803), human hepatocarcinoma (HepG2), and urinary bladder carcinoma (T24) cell lines [10]. Compound **2** was further tested on human lung adenocarcinoma (A549) and drug-resistant lung-cancer cell lines (A549/CDDP). Doxorubicin was used as a positive control. The cell lines were purchased from the Shanghai Cell Bank of the Chinese Academy of Sciences. In short, 1.0 × 10^5^ cells per well (in DMEM with 10% fetal bovine serum) were individually cultured in 96-well microtiter plates. Plates were incubated in a humidified atmosphere with 5% CO_2_ at 37 °C overnight. Cells were treated in triplicate with five concentrations (2.5, 5, 10, 20, and 50 μM) of the tested compounds and doxorubicin at 37 °C for 48 h. Cells were stained with 10 μL (10 mg/mL) of MTT in the incubator (Thermo Fisher, Waltham, USA) at 37 °C for about 4 h. After removal of the supernatant, 100 μL of DMSO (SYCC, Shenyang, China) was added to dissolve the formazan crystals. The absorbance was read using a microplate reader (TECAN, Zürich, Switzerland) at 570/630 nm.

### 4.7. Statistical Analysis

The data were processed with Student’s *t*-test using SPSS software (17.0; IBM^®^, USA), with a significance level of *p* < 0.05. IC_50_ values were determined through a Probit test in SPSS. All the tests were repeated in three independent experiments.

## 5. Conclusions

In conclusion, two oleanane-type triterpenoid saponins, foegraecumoside O (**1**) and foegraecumoside P (**2**), with unique 4/5/6 fused tricyclic skeletons, were isolated from *L. foenum-graecum*. They represent the very first example of oleanane-type triterpenoid saponins bearing a ring linked between C18 and C30. This discovery expands the structural diversity of the oleanane-type triterpenoid saponins. Furthermore, these two compounds are epimers that have opposite configurations of 30-OH. Interestingly, only two with 30α-OH were found to be active to cancer cell lines NCI-H460, MGC-803, HepG2, T24, A549, and A549/CDDP. It provides a potential lead molecule in drug development for treating diseases related to cancer and drug resistance to cancer.

## Data Availability

Not applicable.

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
