# Peer review of "Foegraecumoside O and P, a Pair of Triterpenoid Saponins with a 4/5/6 Fused Tricyclic Oleanane Carbon Skeleton from Lysimachia foenum-graecum Hance"

_molecules, 2023, doi:10.3390/molecules28135061_

Round 1
Reviewer 1 Report
The MS entitled ``Foegraecumoside O and P, a Pair of Triterpenoid Saponins with a 4/5/6 Fused Tricyclic Oleanane Carbon Skeleton from Lysimachia foenum-graecum`` by Dai et al. described the isolation of a pair of 18 oleanane-type triterpenoid saponins with an unprecedented 4/5/6 fused tricyclic skeleton that were determined by chemical methods, extensive spectroscopic analyses, along with quantum chemical calculations. Also the cytotoxic activity as well as the biosynthesis of these compounds were listed.
This manuscript cannot be published in its current from the following issues raised below should be carefully considered.
English editing is needed there are many grammatical and typing mistakes throughout the whole MS.
The plant family name should be added in the abstract and keywords.
In the abstract,
Triterpenoid saponins have been found as the main components., Main components of what?
Replace the verb discovered by isolated. Add the plant part and extract from which these compounds were isolated.
For the compounds name, use small first letter.
Add the type of assay, IC50 values as a range and results of positive control.
Add triterpenoids to the keywords.
Introduction is so short, authors can write about the cancer specially the lung cancer, prevalence and treatment, what are the drawbacks of current treatment and what is the need to discover new anticancer agents.
A reference at the end of line 28 and 29 should be added.
Authors mentioned the uses of the studied plant, in which traditional medicine are this plant used. What are the reported constituents from this plant and their bioactivity if present.
A brief paragraph, as compounds purification should be added at the beginning of the results and discussion section.
Avoid the repeating of the whole name of the compounds, just use compound 1 or compound 2, just the number.
The HMBC correlations that supported the assignment of the methyl signals, and their location should be discussed. Also, the same for the oxymethine and oxymethylene groups should be discussed.
Check all 1H and 13 C NMR chemical shift values in the text with those in the table.
The significant peaks in the IR data should be discussed in the compound discussion section.
Before the biology part in the results and discussion
Add subsection Cytotoxic activity.
The value of the RF in the table differs from that in the text., Revise.
English editing is needed there are many grammatical and typing mistakes throughout the whole MS.
Reviewer 2 Report
General comments
The paper titled: Foegraecumoside O and P, a Pair of Triterpenoid Saponins with a 4/5/6 Fused Tricyclic Oleanane Carbon Skeleton from Lysimachia foenum-graecum has originality on the part of the authors and is complies with the Scope of the journal. However, some shortcomings should be resolved before recommending this article for publication.
I feel that the paper should be rephrased, I detect high similarity with the previous study published by the authors https://doi.org/10.1016/j.phytochem.2017.01.021.
MINOR REVISIONS
I recommend that the authors add a graphical abstract.
Title: Add the name of the taxonomist in the title: Lysimachia foenum-graecum Hance.
Affiliation: I recommend that the authors use their institutional e-mail addresses.
Abstract: The abstract is concise; however, it is important to mention relevant results against tumor cell lines in the abstract.
Keywords: Keywords should be changed and use different ones as in the title.
Introduction: To mention a little about the importance of this plant species. In line 35 mention some examples of which cell types have shown effectiveness or cytotoxicity. Also, in line 36 please mention some examples of the compounds they have identified in their previous research.
Line 44: Change anticancer activity to growth inhibitory effect on tumor cells. You did not perform in vivo assays, only in vitro, and you did not determine the cellular death mechanisms caused by the compounds. Therefore, it cannot yet be asserted that the molecules have antitumor capacity in vivo.
MAJOR REVISIONS
Results section
Throughout the manuscript, m/z should be in italics.
Sometimes 1H NMR, 13C NMR is shown, and sometimes 1H-NMR, 13C-NMR. Please standardize.
Figures 4 and 5: If it is possible to improve the axes of the figures, they are not understandable.
Tables 2 and 3: Adding figure captions. Indicate if it is the mean plus standard deviation as well as the statistical analysis used. Rearrange the table of results and add the IC50s, SDs, as well as the lower and upper limits of the analysis. Also, could you add the results of each concentration evaluated and perform a post-hoc analysis (Tukey) to determine the differences between the groups?
It is important to add assay results in healthy cells (e.g., VERO and PBMC), and to determine the selectivity indices. This to determine the degree of toxicity and selectivity.
It is recommended to determine more assays focused on toxicity or determination of molecular mechanisms of action, e.g., caspases induction, and nitric oxide, among others. Check: PMID: 34579443, PMID: 19235586
Crude extracts and/or partitions should be evaluated against cells as well. This is to give a broad spectrum of cytotoxic activity.
Discussion section:
The discussion should be improved.
The discussion section should be divided into individual one.
Your results should be discussed against previous or similar research.
Discuss the importance of Foegraecumoside molecules, including your previous research (PMID: 28173950).
You need to discuss the results of cytotoxic activity. They also do not discuss the importance of the resistant factor (RF); why is this important? As in your previous paper PMID: 28173950.
Materials and Methods
I suggest attaching a Reagents section where the brand and manufacturer are indicated.
3.1. General Experimental Procedures: Indicate the country of origin of the equipment used. In line 164, I think it should be a spot, not sport.
3.2. Plant Material: Indicate the date of collection and geographic coordinates. Please check the following links to verify the taxonomy:
https://powo.science.kew.org/taxon/urn:lsid:ipni.org:names:701126-1 , http://apps.kew.org/herbcat/getHomePageResults.do?homePageSearchText=Lysimachia+foenum-graecum , http://www.theplantlist.org/tpl1.1/record/tro-26400791
3.3. Extraction and Isolation: Lines 174 to 176, Indicate how long the extractions were made and if Soxhlet equipment and temperatures were used. Indicate if a rotary evaporator was used and the temperatures in the water bath, if applicable, as well as the manufacturer's data.
I also recommend adding a diagram of the bidirected isolation where it indicates the extraction percentages in each phase.
3.4. Acid Hydrolysis: Add missing manufacturer data.
3.3. Cytotoxicity Assays: In line 215 3-(4,5-dimethylthiazol-2-yl)-2,5-diphenyltetrazolium bromide (MTT) reduction assay. Line 216 Indicate the cell type or origin, e.g., HEP-G2 (human hepatocarcinoma). Check: PMID: 37299182., PMID: 37109486
Line 221.- Brand of incubator and microplate, also DMSO manufacturer.
Other comments in the Methodology section:
The results section should be improved and should be divided into sections; sometimes it is confusing.
Add an individual section indicating the statistical analyses performed. Fix the SPSS data; the developer is IBM®.
How were the IC50s determined? Was it by the Probit test?
Conclusion section
The conclusion should be improved, indicating against which cells the cytotoxic effect is seen, as well as adding future perspectives.
SUPPLEMENTARY DATA
I recommend the use of the Journal's template. The supplementary material is consistent, and the images are of high quality. Only a few changes. Also, I recommend that the authors use their institutional e-mail addresses.
Title: Add the name of the taxonomist in the title: Lysimachia foenum-graecum Hance.
Table of Contents (Page 2): Sometimes 1H NMR, 13C NMR is shown, and sometimes 1H-NMR, 13C-NMR. Please standardize.
Figures: Please homogenize the figure captions. The captions of the figures are too close to the following figures. Please put a space between each figure; sometimes, it is confusing. Mark in bold the word Figure in each of the figure captions.
I suggest splitting between Compound 1 and Compound 2. I also suggest splitting between each spectroscopic analyses. Split between IR, 1H-NMR, 13C-NMR, and two-dimensional spectroscopy (COSY, HMBC, …..NOESY). Also indicate the expansion area of each figure e.g., Figure S30. 13C-NMR assignment 2 of compound 2. Expansion in the area from 16 - 18 ppm.
Reference Section
References must conform to the journal's standards, scientific names and words derived from Latin should be in italics.
Reviewer 3 Report
The manuscript presents the isolation, structural characterization and proposes a biosynthetic pathway for two oleanane-type saponins Foegraecumoside O and Foegraecumoside P. Also, the authors had tested the isolated compounds on different cell lines. I consider that the manuscript is well structured, the methods are well described, as well as the results. The supplementary data offers the adequate information that supports the results presented in the manuscript. Particularly interesting, is the structure of these saponins, with a ring linked between C18 and C30.
I suggest the acceptance of the manuscript in its present form.
Round 2
Reviewer 1 Report
The quality of the MS has been greatly improved after carrying out the required corrections.
Reviewer 2 Report
All the comments are addressed.